# Update on Effects of the Prophylactic HPV Vaccines on HPV Type Prevalence and Cervical Pathology

**DOI:** 10.3390/v16081245

**Published:** 2024-08-02

**Authors:** Ian N. Hampson, Anthony W. Oliver

**Affiliations:** 1Division of Cancer Sciences, University of Manchester, Oxford Rd, Manchester M13 9WL, UK; 2Ravan Bio Ltd., Unit 7A, Kilburn House, Lloyd St North, Manchester M15 6SE, UK; awo@ravanbio.com

**Keywords:** HPV, vaccines, prophylactic, invasive cervical cancer, HPV type-replacement, clinical unmasking, cervical intraepithelial neoplasia, CIN, superinfection exclusion

## Abstract

Most national prophylactic HPV vaccination programs started in approximately 2008, with either the bivalent Cervarix HPV16/18 or quadrivalent Gardasil (HPV6/11/16/18) vaccines, which were then followed by introduction of the nonavalent Gardasil 9 (HPV6/11/16/18/ 31/33/45/52/58) vaccine from 2015. Since that time, these products have demonstrated their ability to prevent infection with vaccine-covered HPV types and subsequent development of HPV-related cervical and genital pathologies. The data indicate that vaccination of young girls prior to sexual debut is more effective than vaccination of older HPV+ve women. Although some studies have shown a decline in the prevalence of vaccine-covered HPV types, there are national and regional differences in overall vaccine efficacy. Furthermore, several recently published studies show an increase in the prevalence of non-vaccine-covered HPV types in vaccinated populations, which is indicative of HPV type-replacement. It is also notable that vaccine-related changes in HPV type prevalence spread between vaccinated and unvaccinated women at the same geographical location—presumably via sexual transmission. In conclusion, it is not yet clear what effect dissemination of vaccine-associated changes in HPV type prevalence will have on vaccine efficacy and cervical pathology, particularly in mixed populations of vaccinated and unvaccinated women. However, it is very clear these observations do underscore the need for long-term continuation of cervical screening combined with regular reassessment of testing practices.

## 1. Introduction

The following discussion provides an updated perspective on previously reported effects of the prophylactic HPV vaccines on HPV type prevalence and cervical pathology [1].

Although there have been some concerns over vaccine-related adverse events [2,3], these have not established causality and both Cervarix and Gardasil are now judged to be well tolerated and safe [4]. Moreover, there are now many studies which have shown that prophylactic vaccination with either product is effective when given to young women prior to sexual debut and HPV infection [5]. This was demonstrated for Cervarix in a trial carried out in Costa Rica [6] although non-HPV16/18 types were still present in vaccinated women at the same location [7]. Indeed, all the current vaccines are effective against specific HPV types and they have the potential to reduce the global incidence of cervical cancer, albeit this will be over a period of 30–40 years from the onset of vaccination. However, it is also very clear that extensive cervical screening, combined with treatment of detected disease, will still be an absolute necessity during this period, particularly for older unvaccinated women [8]. Moreover, the overall long-term impact of vaccination will depend to what extent this is influenced by other factors such as HPV type-replacement [9,10] and superinfection exclusion [11].

To clarify these last two points, it is important to realise there are >50 different types of HPV known to infect the genital epithelium. These consist of ~14 well established high-risk (HR) types in addition to ~36 types which are either probable high-risk or low-risk (LR) types. Most significantly, it is well known that HPV infections of one type can affect susceptibility to infection with others by either stimulating cross-type immunity [9] or by superinfection exclusion [11]. In this way, the cervix may play host to a changing population of different HPV types. Given this level of complexity, it is not clear what effect the HPV vaccines will have on the prevalence of non-vaccine-covered HPV types and, in the longer term, any associated dysplasia. In order to address this issue, there are important questions which remain to be answered. 

Are infections with vaccine-covered high-risk HPV types replaced by HPV types which are not covered by the vaccines and how does this affect the incidence of cervical intraepithelial neoplasia (CIN) and invasive cervical cancer (ICC)? Are vaccine-induced changes in HPV types that are observed in vaccinated women then transferred to unvaccinated women in the same population? 

## 2. Post-Vaccination Changes in HPV Type Prevalence in Screening Populations

Evidence supporting vaccine-related HPV type-replacement is provided by a cross sectional study of two Spanish communities vaccinated with either Cervarix or Gardasil [12]. Analysis of pre and post-vaccination prevalence of 35 HPV types showed that LR HPV6/11 and HPV16 infections declined, although the latter was not statistically significant and there was no change in the prevalence of HPV18. Furthermore, Gardasil does not cover HR types 31, 52 and 45, which all showed a significant post-vaccination increase in prevalence. Most notably, Gardasil 9 does not cover HR HPV types 35, 39, 56, 59 and 68, which also increased post-vaccination (see Supplementary Material, additional data file in [12]). These findings are consistent with studies carried out in New York (see Figures 1 and 2 in [13]) and Stockholm (see Figure 1 in [14]), which shows a post-vaccination increase in non-vaccine-covered high-risk HPV types in both vaccinated and unvaccinated women. 

In 2010, the prevalence of HR and LR HPV types was evaluated in cervical smears taken from 3817 women aged between 25 and 64 from multiple screening centres in Central and Southern Italy [15]. This was followed by a larger Italian multi-centre study in 2013 which analysed smear samples from 37,077 unvaccinated women aged 25–60 years, for the prevalence of multiple HR HPV types in order to provide baseline data for vaccine efficacy [16]. In 2023, following a 14-year vaccination period, analysis of multiple HR HPV types was carried out on smear samples taken from 30,445 from women aged 30–64 years from the Lazio region of Italy [17]. Comparison of these three studies showed the number of women positive for any of 13 HR HPV types was 9% in 2010 [15], 8% in 2013) [16] and 13.9% (2023) [17], which equates to an average increase of ~65% between 2010 and 2023, as illustrated in Figure 1A. Analysis of the prevalence of individual HR HPV types found by these three studies is shown in Figure 1B. It is surprising that, in spite of 14 years post-vaccination, there was no overall decrease in the percentage of HPV 16- or 18-positive women, and all the other HR HPV types analysed actually showed a marked increase in the 2023 population. Moreover, it is also very clear that the observed increase in prevalence of non-vaccine-covered HPV types is supportive of vaccine-related type replacement. 

An alternative explanation for these findings could be related to differences in sensitivity of the different HPV detection systems used. However, comparison of Hybrid Capture 2 (as used in 2010 and 2013) with Anyplex™ II HPV HR (As used in 2023) has shown their sensitivities to be very similar [18]. Another confounding factor could be related to differences in the age distribution of study participants where younger women are known to have higher rates of HPV infection than older women. This was demonstrated by Rossi et al. [15] but not seen by Carozzi et al. [16], who both analysed samples from women aged ≥25 years. Thus, it is significant that the 2023 study of Cenci et al. [17] only enrolled women aged ≥30 years, which could have potentially had a negative impact on HPV prevalence rather than the observed increase.

**Figure 1 viruses-16-01245-f001:**
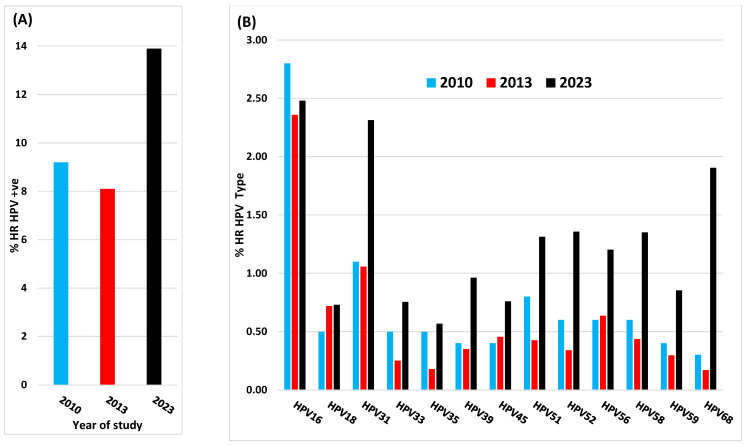
Prevalence of HR-HPV in Cervical Screening Populations in Italy from 2010 Giorgi Rossi et al. [15], 2013 Carozzi et al. [16] and 2013 Cenci et al. [17]. (**A**) Total percentage infection for all HR HPV types; (**B**) Percentage of infections with individual HR HPV types.

Even though HPV vaccine coverage in Italy has been below the desired target, particularly during the COVID pandemic, for many years it still achieved >70% [19]. Thus, the lack of any post-vaccination reduction in the prevalence of HPV16 and 18 in 2023, combined with an increase in non-vaccine-covered HPV types, is counterintuitive. It could be argued that any effects of vaccination on HPV type prevalence may be diluted by the inclusion of older unvaccinated women although, as previously discussed, it has been shown that vaccine-related changes in HPV type prevalence spread between vaccinated and unvaccinated individuals at the same geographical location [13,14]. It is notable that Pisani and Cenci 2024 also analysed the prevalence of infections with multiple HPV types in individual women [20] using the same patient cohort as their earlier study [17]. Had they been available, analysis of this feature in samples from the same location, prior to the onset of vaccination, would have proved informative. 

## 3. Post-Vaccination Changes in HPV Type Prevalence and/or Associated Cervical Dysplasia

Section 2 discussed the effects of vaccination on overall HPV type prevalence in smears from women invited for routine screening irrespective of cytology or pathology. However, to determine the impact of vaccination on the development of CIN, it is critical to analyse HPV type prevalence in women with these pathological changes. 

As previously discussed [1], an observational study from Northern Italy assessed the impact of vaccination between 2005 and 2019 on HPV prevalence in cervical smears and colposcopy-directed biopsies taken from 5864 women aged between 21 and 65 [21]. Comparison of cervical smears from 2005 to those taken in 2019 found that the incidence of LR HPV types showed a ~50% reduction whereas Gardasil 9-covered HR HPV types showed little change over the study period. However, the prevalence of non-vaccine-covered HR HPV types and untypable HPVs had both increased >100% in 2019 (see Table 2 in [21]). A reduction in the incidence of HPV16 and HPV31 was found in biopsy-confirmed CIN1 but only in younger women aged 21–29 years. This was not seen in women aged ≥30 years and no reduction in HPV16 incidence was observed for women with CIN2. Most notably, an increase in HPV-negative lesions, those positive for non-vaccine-covered HPV, and unknown HPV types was also seen across all age groups. These results support vaccine-related HPV type replacement and the observation that the majority of lesions remained positive for all HR HPVs targeted by the vaccines, irrespective of age or severity of cervical lesion, is cause for concern.

Gardasil was the main vaccine used in the previously discussed observational Italian study [21]. A more recent report examined long-term follow-up of women treated with Cervarix in the double-blinded Costa Rican HPV Vaccine Trial (CVT) [22]. Initially 7466 women aged 18–25 were randomly assigned either Cervarix or a hepatitis A control vaccine and both arms followed up for a period of 1–4 years. After the follow-up period, women in the hepatitis A control arm were then offered vaccination with Cervarix. At this time an observational unvaccinated control group (UCG) of 2836 women from the same birth cohort and geographical location as the previous vaccine arms was recruited. The incidence of cervical dysplasia and HPV type in the original Cervarix arm and the more recent UCG arm was then evaluated over a further period of 7 years, providing a total of 11 years of follow-up. Most notably, this study specifically addressed the incidence of CIN related to infection with non-vaccine-covered HPV types. At years 1–4, a ~50% reduction in HPV16/18-related CIN2 and CIN3 was found in the Cervarix arm compared to the hepatitis A vaccine control arm. Furthermore, when compared to the UCG at years 7–11, this had reduced by 90% for CIN2 and 87% for CIN3. Significant cross-protection against HPV31/33/45-related CIN2/3 was also observed for both time periods, whereas protection against strain types covered by Cervarix vaccination was as expected. However, a marked increase in the incidence of CIN positivity for non-vaccine-covered HPV types 35/39/51/52/56/58/59 was found in Cervarix-vaccinated women. For CIN2 at years 1–4, there was no change, whereas at years 7–11 this had increased by 71.2%. For CIN3 at years 1–4, there was an increase of 17.4%, but at years 7–11 this had increased to 135%. 

The CVT follow-up data reported by Shing et al. [22] provides strong evidence in support of vaccine-induced HPV type replacement in cervical lesions, although these authors propose the alternative explanation of “clinical unmasking” for this effect. This is thought to be due to a reduction in the numbers of surgical procedures carried out on vaccinated women for lesions positive for vaccine-covered HPV types (e.g. 16/18). Since a greater number of cervical transformation zones will remain intact in this population, this leads to “clinical unmasking” or revealing of any non-vaccine-covered HPV types which may be present. Instead of being removed by surgery, any non-vaccine-covered HPVs will remain in place to induce subsequent neoplastic changes. Whether it is HPV type replacement or clinical unmasking driving the observed increase in CIN lesions positive for non-vaccine-covered HPVs with either Cervarix- or Gardasil-vaccinated women, the outcome is still the same.

Although the CVT study by Shing et al. [22] and the Italian study by Gardella et al. [21] used different HPV vaccines and age groups, both reported a reduction in lesions positive for vaccine-covered HPV types in younger women. Additionally, both studies observed a very significant increase in lesions positive for non-vaccine HPV types. Similar results were also reported in a study by Cuschieri et al., who analysed HPV type prevalence in biopsies from >1700 Scottish women aged 20–25 diagnosed with CIN between 2011 and 2018 [23]. Vaccination of girls aged 12–13 started in Scotland with Cervarix in 2008 and has had a consistently high 80% uptake. Analysis of 24 HPV types in CIN2 lesions from vaccinated women showed ~100% reduction in lesions positive for types 16/18 when compared to unvaccinated women. Comparison of vaccine cross-protected HPV types in CIN2 showed vaccination-associated reduction in the prevalence of types 31/45, although type 33 had increased by 100%. For CIN3, vaccine-associated reduction of type 16/18-positive lesions was less pronounced and type 33 lesions had also increased by >100% and, unlike CIN2, there was no reduction in prevalence of type 31 lesions. Furthermore, there was also a marked increase in non-vaccine-covered HPV types 52/51/33/58/39/82/66 in CIN2/3 from vaccinated versus unvaccinated women (see Figure 2 in [23]). In order to clarify the overall impact of vaccination on the prevalence of HPV types found in CIN2/3, Figure 2 and Figure 3 show these results reformatted to illustrate the percentage of vaccine-related change in lesions positive for vaccine-covered versus non-vaccine-covered HPV types. Overall vaccination induced a significant reduction in the prevalence of HPV16/18-positive CIN2/3, although there was clearly a marked increase in lesions positive for other HR HPV types.

**Figure 2 viruses-16-01245-f002:**
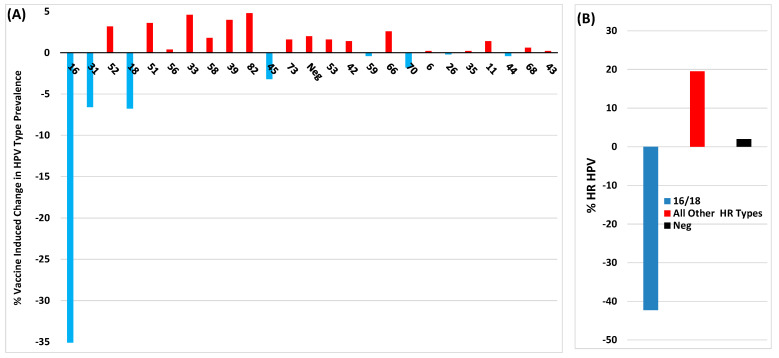
Percentage difference in HPV types found in CIN2 lesions between vaccinated and unvaccinated young Scottish girls (Data extracted from Cuschieri et al. [23]). (**A**) Percentage difference between lesions +ve for individual HPV types. (**B**) Percentage difference between lesions +ve for HPV16/18 versus the sum of the difference between +ve and −ve values for all the other HPV types tested.

**Figure 3 viruses-16-01245-f003:**
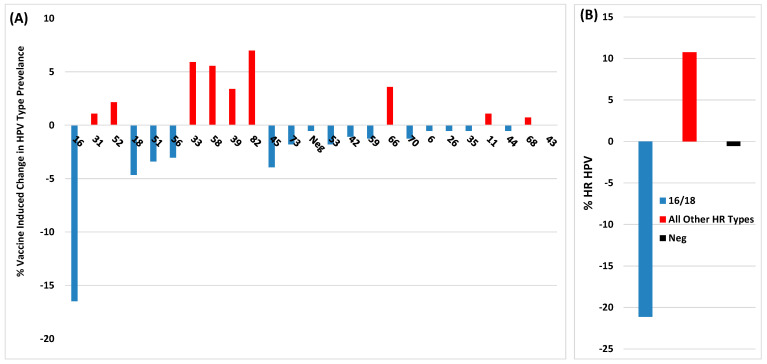
Percentage difference in HPV types found in CIN3 lesions between vaccinated and unvaccinated young Scottish girls (Data extracted from Cuschieri et al. [23]). (**A**) Percentage difference between lesions +ve for individual HPV types. (**B**) Percentage difference between lesions +ve for HPV16/18 versus the sum of the difference between +ve and −ve values for all the other HPV types tested.

The previously discussed findings are also consistent with an American study which investigated changes in CIN prevalence across five states between 2008 and 2015 [24]. HPV vaccination was approved in the USA in 2006 although uptake was slow, achieving only 40% in 2014 [25]. Analysis of the incidence of 16,572 CIN2 lesions in American women aged 18–39 years demonstrated a reduction in the prevalence of CIN2 for women aged 18–24. However, a marked increase was seen in women aged 25–39 years with the same trend observed for CIN3, although HPV types per lesion were not specified [24].

## 4. Post-Vaccination Changes in HPV Type Prevalence and/or the Incidence of Cervical Cancer (ICC)

CIN precedes ICC, which typically takes 10–20 years to develop and, as stated, HPV vaccination started in the USA in 2006. Using CIN as a proxy for subsequent development of ICC, the previously discussed work has consistently shown that vaccination is most effective when given to young girls aged 10–13 prior to sexual contact. 

When vaccination was first introduced in the USA, it was given to females 9–26 years old with the most uptake occurring in older age groups until ~2011, when the situation reversed and younger girls aged 11–13 became the main recipients (see Figure 1A in [25]). Even so, uptake in the US has been poor, with estimates indicating that only 40% of 13-year-old girls were vaccinated in 2014 and 2016 [25].

A study from the United States Cancer Statistics (USCS) database assessed age-related changes in the incidence of squamous cell carcinoma (SCC) of the cervix in women aged 15–29 years between 1999 and 2017 [26]. A total of 13,231 cases of ICC were stratified for incidence/year in women aged 15–20, 21–24 and 25–29 years. In women aged 15–20, a total of 193 cases of SCC were diagnosed over the entire study period and a reduction in the incidence rate/year was observed from 2012 onwards (Figure 2 and Table 2 in [26]). A total of 1491 cases of SCC were diagnosed in women aged 21–24 years, which showed a reduction in incidence rate/year of ~0.4 from 1999 to 2011 and a further decrease of ~0.5 from 2012 to 2017 (Figure 3 and Table 2 in [26]). A total of 7246 cases of SCC were diagnosed in women aged 25–29 years, which showed a pronounced drop of ~2.0 in incidence rate/year from 1999 to 2011. However, most notably, from 2012 to 2017, the incidence rate per year in this age group actually increased by ~0.7 (Figure 4 and Table 2 in [26]), which is consistent with the previously discussed increased prevalence of CIN2/3 seen in American women over 25 between 2008 and 2015 [24].

Given the initial older age of vaccine recipients, combined with low uptake in the USA [25], it is very unlikely that the observed drop in SCC incidence rate/year for women aged 21–29 years between 1999 and 2011 [26] has any relation to vaccination and is probably due to changes in screening practice (see Figure 1 in [26]). It is much more likely that the increased drop in incidence rate/year seen for women aged 15–24 years from 2012 to 2017 could be related to the effects of vaccination. However, what is the explanation for the observed increase in incidence rate/year in women aged 25–29 during the same study period? Since vaccination uptake in the USA was initially greatest in 16–19-year-olds in 2007 [25], it is very clear that a significant proportion of these will be captured in the 2012–2017 USCS cohort of women aged 25–29 years [26]. As a consequence, it is highly likely this cohort will also contain the largest number of women who were HPV-positive at the time of vaccination when it proves less effective. However, this still does not explain the increase in SCC incidence rate/year between 2012 and 2017 and it is also notable that these data do not include the age group with the highest incidence of SCC in the USA, which is 30–44 years.

A very recent study extracted historical data in July 2020 from the Scottish Cancer Registry population to analyse the effects of HPV vaccination on the incidence of ICC in Scotland [27]. An unvaccinated cohort of 294,221 girls born between 1988 and 1996 was first identified from the Scottish Cervical Cancer Call/Recall System. Vaccination of girls aged 14–18 was given from 2008 to 2009 followed by routine vaccination of 12–13-year-olds continuing from 2008 onwards. In total, 239 cases of ICC were identified, which were then stratified by vaccination status and age group. Although a much smaller number of ICC cases were used than in the preceding American study [26], vaccination status per individual was known and vaccine uptake was higher, achieving 80% in Scotland. Compared to an incidence rate of 8.4 cases/100,000 women/year in the control unvaccinated cohort, no cases of ICC were detected in girls vaccinated when aged 12–13 years, although this group was tenfold smaller than the unvaccinated control group. Nevertheless, a much larger group of girls vaccinated at ≥14 years also had a reduced incidence of 3.2 cases per 100,000 women/year. However, the incidence rates/100,000 women per year for age groups 14–16, 17–18 and >18 are not shown, although the results required to calculate this are included (see Table 2, footnote “d” in [27]). Notably, 3601 women were vaccinated at >18 years, of whom <5 developed ICC. If the same type of calculation is performed, as for the younger age groups, these data do indicate a higher incidence rate for these women than the 8.4 cases/100,000 women/year in the unvaccinated control group, which is consistent with many of the preceding cited studies.

These aforementioned studies demonstrate a clear underlying trend of age group-specific changes in the incidence rate of ICC in HPV vaccinated communities as a general phenomenon which is not confined to a single country. For example, in 2021 Swedish women aged 15–24 years had an all-time low incidence of ICC at 0.52/100,000 women/year whereas women aged 25–39 had an incidence rate of ~15/100,000 women/year, which showed little change from 1977–2007. Indeed, from 2008, the rate increased for women aged 25–44, peaking at 20.5/100,000 in 2014, which then reduced to 16.1/100,000 in 2021 (see hyperlink reference [28], accessed 17 June 2024). However, this rate is higher than that found for the same age group in 1977, despite intervening improvements in screening technology combined with the onset of vaccination. Arguably, out of all the Nordic countries, Finland has seen the most profound changes in ICC incidence over time. In 1965, prior to the onset of cervical screening, the incidence of ICC in Finnish women aged 25–44 was 17.2/100,000. By 1985, this reduced to 2.9/100,000 which was one of the lowest incidence rates in the world at that time. Finland introduced a national HPV vaccination program for young girls in 2013 and yet, irrespective of screening and vaccination, from 1990 onwards the rate of ICC has steadily increased until the current peak rate of 15.8/100,000 in 2021 (see hyperlink reference [29], accessed 17 June 2024). Furthermore, it is significant that a similar trend of increased incidence rates in women aged 25–34 years has also been observed in the UK, where this has increased from 13/100,000 in 2004 to 19/100,000 by 2018 (see hyperlink reference [30], accessed 17 June 2024). These UK data also highlight the very low incidence rate of 1 case of ICC per 100,000 for women <24 years old, and which has remained the same since 1993. 

For a previous discussion of national ICC incidence rates over time in selected countries, see Hampson et al. [11]. 

## 5. Potential Economic Implications

Cost estimates of each quality-adjusted life year (QALY) gained from HPV vaccination have been determined from cost-effectiveness analyses (CEA) [31]. However, this has shown large variability depending on age at vaccination. For example, women aged 12 through 26 vaccinated with Gardasil 9 have an estimated QALY of <USD10,000 whereas this increases to USD653,000 for catch-up women vaccinated aged 12 through 46 years. Based on these figures, it was concluded that vaccination of younger women was significantly more cost effective than that of older women, although there are important questions that still need to be addressed in order to improve the accuracy of these estimates [32]. For example:How long will vaccine protection last and what is the extent of cross-type immunity?What is the extent of vaccine-related HPV type replacement?What are the consequences of changes in HPV types spreading between vaccinated and unvaccinated individuals?

Although the previously cited studies have shown some cross-type protection, they have also provided strong evidence supporting HPV type replacement and/or clinical unmasking, though the long-term impact of this is yet to be determined. 

## 6. Conclusions

It should be noted that the preceding analysis depends on the assumption that the prevalence of individual HPV types within specific age groups and geographical locations do not undergo natural variation between different time periods although, with large numbers, this should even out.

It is clear that, when given to young women, prior to sexual debut, the current vaccines have demonstrated very significant efficacy against vaccine-covered HPV types and associated disease. However, prior to vaccination, HPV16 was the most common HR type associated with ICC, but there is now a considerable body of evidence which shows a significant increase in the prevalence of non-vaccine-covered HR types following vaccination [12,13,14,15,16,17,21,22,27]. Furthermore, at this stage it is not clear to what extent these alterations in HPV type prevalence will disseminate between mixed populations of vaccinated and unvaccinated women. Most importantly, the oncogenic potential of these changes has not yet been explored in these populations and it has been speculated they may take significantly longer to cause ICC than HPV16. Indeed, it has been shown that women develop ICC, positive for HPV16, 18 or 45, 10.5 years younger than women with tumours positive for other HPV types [33]. 

Thus, moving forward, it is very clear that extensive long-term surveillance of at least >30 years from the present day will be essential in order to monitor vaccine-related changes in HPV type prevalence and how this affects the subsequent development of neoplasia. Furthermore, it will also be very important to reassess both HPV and cytology testing practices in order to accommodate the impact of vaccine-related changes on the repertoire of HR cervical HPV infections over time [34]. Notably, these and other practices have been previously advocated in our earlier review on the effects of the prophylactic HPV vaccines [11]. 

The previously discussed assessment of national changes in the incidence of ICC, before and after the introduction of vaccination, highlights some interesting observations. For example, the recent increase in the incidence of ICC for women aged >25 in the Nordic countries [28,29] and the UK [30] is paradoxical considering their extensive vaccination programs, with a high uptake >80%. Obviously, it may be too early for vaccination to have impacted on these statistics, although these findings clearly warrant further investigation, as indicated in our previous review [11].

Although the current update has focused on the effects of the prophylactic HPV vaccines on the development of cervical cancer, it is very clear they also have the potential to influence the development of HPV-related head and neck cancers. In this regard, it is notable that comparison between oral and genital sites has shown that HPV16 has a significantly higher prevalence in the oral cavity than other HPV types [35]. This, in turn, reflects the observation that 85–96% of all HPV-related oropharyngeal cancers (OPCs) are positive for type 16 and they also occur at an earlier age than HPV-ve oral tumours [36]. Thus, it is possible the high dependence of OPC on HPV16 may reduce any potential carcinogenic effects of vaccine-related HPV type replacement in the oral cavity. Furthermore, OPC is more common in males than females, which clearly supports the prophylactic vaccination of boys. However, since this was introduced approximately 10 years later than for girls, it will take at least 20–30 years for this to have a direct effect, although indirect herd effects reducing transmission of HPV16 from vaccinated girls are already apparent [36]. 

As a final point, it is has been shown that Gardasil has efficacy as an adjuvant therapy against benign HPV6/11-related recurrent respiratory papillomatosis (RRP) which, given its prophylactic mode of action, may seem counterintuitive [37]. However, the most likely explanation is that, unlike high-grade CIN, RRP is a productive HPV infection which still expresses the L1 vaccine target protein, which promotes post-surgical immune surveillance of regrowing RRP lesions.

## Data Availability

Not applicable.

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
