# Peer review of "Update on Effects of the Prophylactic HPV Vaccines on HPV Type Prevalence and Cervical Pathology"

_viruses, 2024, doi:10.3390/v16081245_

Round 1

Reviewer 1 Report

Comments and Suggestions for Authors

Dear Professor Hampson,

Thank you for the opportunity to review your work. You pose an intellectually provoking question about the rise of non-vaccine HR HPV types. This is both interesting and concerning and does justify publication. You are aware that the prevalence of HPV does vary according to geography and age of patient on cervical smear testing. What your work cannot be certain of is any skew in each age category. i.e. is the mode of age the same between different time periods for a particular age category. I agree that with large numbers this should even out but it is a consideration.

Also, I understand the that non-vaccine HPV types increase in vaccinated individuals as we might expect but the impact of this is uncertain and I feel that discussing this would be of greater direct value. i.e. how bad are the replacement HR HPV with respect to prognosis?

Furthermore, this work concentrates, quite reasonably, on cervical disease but a wider understanding of HPV disease may offer more to the reader. Mention of impact of HPV driven Head & Neck cancer and recurrent respiratory papillomatosis is important, especially the latter where HPV vaccine is being used as a therapeutic and seems to have some utility perhaps more in children than adults with this disease.  

Author Response

Authors: We are very grateful to the reviewer for taking the time to review our manuscript and for their valubale suggestions which we have acted on as follows:

Reviewer: What your work cannot be certain of is any skew in each age category. i.e. is the mode of age the same between different time periods for a particular age category. I agree that with large numbers this should even out but it is a consideration.

Authors: The phrase " on the assumption that the prevalence of individual HPV types within specific age groups and geographical locations do not undergo natural variation between different time periods although, with large numbers, this should even out." has been added to section 6, paragraph 1.

Reviewer: Also, I understand the that non-vaccine HPV types increase in vaccinated individuals as we might expect but the impact of this is uncertain and I feel that discussing this would be of greater direct value. i.e. how bad are the replacement HR HPV with respect to prognosis?

Authors. The sentence "Indeed, it has been shown that women develop ICC, positive for HPV16,18 or 45, 10.5 years younger than women with tumours positive for other HPV types" has been added in Section 6, paragraph 2

Reviewer: Furthermore, this work concentrates, quite reasonably, on cervical disease but a wider understanding of HPV disease may offer more to the reader. Mention of impact of HPV driven Head & Neck cancer and recurrent respiratory papillomatosis is important, especially the latter where HPV vaccine is being used as a therapeutic and seems to have some utility perhaps more in children than adults with this disease.

Author: The following has been added at the edn of Section 6.

"Although the current update has focused on the effects of the prophylactic HPV vaccines on the development of cervical cancer, it is very clear they also have the potential to influence the development of HPV-related head and neck cancers. In this regard, it is notable that comparison between oral and genital sites has shown that HP16 has a significantly higher prevalence in the oral cavity than other HPV types [35]. This, in turn, reflects the observation that 85-96% of all HPV related oropharyngeal cancers (OPC’s) are positive for type 16 and also occur at an earlier age than HPV negative oral tumours [36]. Thus, it is possible the high dependence of OPC on HPV16 may reduce any potential carcinogenic effects of vaccine-related HPV type-replacement in the oral cavity. Furthermore, OPC is more common in males than females, which clearly supports the prophylactic vaccination of boys. However, since this was introduced approximately 10 years later than for girls, it will take at least 20-30 years for this to have a direct effect although indirect herd effects reducing transmission of HPV16 from vaccinated girls, are already apparent [36].

As a final point, it is has been shown that Gardasil has efficacy as an adjuvant therapy against benign HPV6/11 related recurrent respiratory papillomatosis (RRP) which, given its prophylactic mode-of-action, may seem counterintuitive [37]. However, the most likely explanation is that, unlike high-grade CIN, RRP is a productive HPV infection which stiil expresses the L1 vaccine target protein which promotes post surgical immunosurveillance of regrowing RRP lesions.

Reviewer 2 Report

Comments and Suggestions for Authors

In this broad-scoping “Perspective’ article the authors examine the effect of the introduction of HPV vaccines on the prevalence of non-vaccine covered HPV types in vaccinated populations and HPV type-replacement. The paper covers vaccine-related changes in HPV-type prevalence spread between vaccinated and unvaccinated women at various geographical locations. It explores the effect of vaccination and age of immunization on the rates of CIN and ICC in vaccinated and unvaccinated women.

This is an extremely well-written review and provides an excellent summary of the effects of adolescent/women-only HPV vaccination on the epidemiological behaviour of  HPV subtypes and  pathological outcomes in women. The paper limits its discussion to the vaccination of women. Other vaccination strategies could be mentioned (even briefly) to give a deeper perspective on the problem. The vaccination of boys is a case in point. (see https://www.bmj.com/bmj/section-pdf/186442?path=/bmj/340/7737/Head_to_Head.full.pdf).

Author Response

Authors: We than the reviewer for their kind comments, for taking the time to review our mansucript and for their valuable suggestions which we have acted on as follows: 

Reviewer: This is an extremely well-written review and provides an excellent summary of the effects of adolescent/women-only HPV vaccination on the epidemiological behaviour of  HPV subtypes and  pathological outcomes in women. The paper limits its discussion to the vaccination of women. Other vaccination strategies could be mentioned (even briefly) to give a deeper perspective on the problem. The vaccination of boys is a case in point. (see https://www.bmj.com/bmj/section-pdf/186442?path=/bmj/340/7737/Head_to_Head.full.pdf).

Authors: This has been added at the end of Section 6

"Although the current update has focused on the effects of the prophylactic HPV vaccines on the development of cervical cancer, it is very clear they also have the potential to influence the development of HPV-related head and neck cancers. In this regard, it is notable that comparison between oral and genital sites has shown that HP16 has a significantly higher prevalence in the oral cavity than other HPV types [35]. This, in turn, reflects the observation that 85-96% of all HPV related oropharyngeal cancers (OPC’s) are positive for type 16 and also occur at an earlier age than HPV negative oral tumours [36]. Thus, it is possible the high dependence of OPC on HPV16 may reduce any potential carcinogenic effects of vaccine-related HPV type-replacement in the oral cavity. Furthermore, OPC is more common in males than females, which clearly supports the prophylactic vaccination of boys. However, since this was introduced approximately 10 years later than for girls, it will take at least 20-30 years for this to have a direct effect although indirect herd effects reducing transmission of HPV16 from vaccinated girls, are already apparent [36].

As a final point, it is has been shown that Gardasil has efficacy as an adjuvant therapy against benign HPV6/11 related recurrent respiratory papillomatosis (RRP) which, given its prophylactic mode-of-action, may seem counterintuitive [37]. However, the most likely explanation is that, unlike high-grade CIN, RRP is a productive HPV infection which stiil expresses the L1 vaccine target protein which promotes post-surgical immunosurveillance of regrowing RRP lesions"..